# Spin current generation in organic antiferromagnets

Makoto Naka[1], Satoru Hayami[2], Hiroaki Kusunose[3], Yuki Yanagi[4], Yukitoshi Motome[5] & Hitoshi Seo[6,7]

Spin current–a flow of electron spins without a charge current–is an ideal information carrier free from Joule heating for electronic devices. The celebrated spin Hall effect, which arises from the relativistic spin-orbit coupling, enables us to generate and detect spin currents in inorganic materials and semiconductors, taking advantage of their constituent heavy atoms. In contrast, organic materials consisting of molecules with light elements have been believed to be unsuited for spin current generation. Here we show that a class of organic antiferromagnets with checker-plate type molecular arrangements can serve as a spin current generator by applying a thermal gradient or an electric field, even with vanishing spin-orbit coupling. Our findings provide another route to create a spin current distinct from the conventional spin Hall effect and open a new field of spintronics based on organic magnets having advantages of small spin scattering and long lifetime.

[1] Waseda Institute for Advanced Study, Waseda University, Shinjuku, Tokyo 169-8050, Japan. [2] Department of Physics, Hokkaido University, Sapporo, Hokkaido 060-0810, Japan. [3] Department of Physics, Meiji University, Kawasaki, Kanagawa 214-8571, Japan. [4] Institute for Materials Research, Tohoku University, Sendai, Miyagi 980-8577, Japan. [5] Department of Applied Physics, The University of Tokyo, Bunkyo, Tokyo 113-8656, Japan. [6] Condensed Matter Theory Laboratory, RIKEN, Wako, Saitama 351-0198, Japan. [7] Center for Emergent Matter Science (CEMS), RIKEN, Wako, Saitama 351-0198, Japan. Correspondence and requests for materials should be addressed to M.N. (email: naka@aoni.waseda.jp)

Organic metals and semiconductors[1] possess a variety of features not shared by inorganic materials, e.g., light, flexible, and toxic-element-free. They have been rapidly developed over the past decades for use in consumer electronic devices, such as organic transistors, light-emitting diodes, and piezo actuators. These accomplishments, in combination with recent evolutions of inorganic spintronics based on spin current physics, have promoted a new field, i.e., organic spintronics. Now significant efforts are being made to elucidate spin transport phenomena in organic semiconductors[2–4]. However, organic spintronics devices are actually not purely organic but are hybrid with inorganic materials, because the generation of spin current basically requires an inorganic magnetic electrode. In fact, attempts for exploiting organic materials as the spin current generator are quite limited[5,6].

Here, we theoretically propose a microscopic mechanism of spin current generation in organic materials utilizing an archetypal antiferromagnet. Figure 1a provides a schematic illustration of the present spin current generation in the antiferromagnetic (AFM) state, where the up and down spins aligned on the molecular checker plate play a role of a spin-rectifier converting a heat-current driven by a thermal gradient, or an electron current by an electric field, into the spin current. When we rotate the external field with respect to the crystal axes in the two-dimensional plane, the direction of the generated spin current rotates in the opposite direction as shown in Fig. 1b. The directional dependence is strikingly different from the conventional spin Nernst and spin Hall effects[7–12], in which the spin current always flows perpendicular to the field direction. As a platform of this phenomenon, we focus on an organic antiferromagnet $\kappa$-(BEDT-TTF)$_2$Cu[N(CN)$_2$]Cl (abbreviated as $\kappa$-Cl).

## Results

**Crystal structure and model**. $\kappa$-Cl is a well-studied insulator, showing a variety of cooperative phenomena, e.g., AFM ordering, insulator-to-metal transition, and superconductivity, at low temperatures and/or under pressures[13–18]. The crystal structure is composed of an alternate stacking of two-dimensional conducting BEDT-TTF (abbreviated as ET) layers and insulating Cu[N(CN)$_2$]Cl layers. Figure 2a shows the molecular arrangement (called $\kappa$-type) in the conducting layer, where four ET molecules in the unit cell form two kinds of dimers with different

orientations, termed $A$ and $B$, connected by a glide operation (mirror and half translation).

This class of organic materials is known to show a simple electronic structure composed of frontier molecular orbitals[19,20]. In the $\kappa$-type materials, the frontier orbitals in each ET dimer become strongly hybridized by the intra-dimer transfer integral shown in Fig. 2b, and constitute bonding and antibonding orbitals. They result in four bands as there are two dimers in the unit cell: two lower(higher)-energy bands are from the (anti-) bonding orbitals, as shown in Fig. 2c. The system has three electrons per two dimers on average, and hence, the four bands are three-quarter filled.

In the last few decades, extensive studies have been made for understanding the cooperative phenomena in this system[21–23]. Most of them, however, are based on the single-band picture, where the two fully occupied bands are disregarded (see the broken lines in Fig. 2c). This approach is justified in the large dimerization limit[19], where the crystallographic distinction of the $A$ and $B$ dimers is lost. In other words, the glide symmetry in the molecular arrangement in the conducting layer was disregarded in the previous studies. In the following, we will discuss that the breaking of the glide symmetry by the AFM ordering plays an essential role in a peculiar spin current generation.

We investigate electronic structures and spin current transport properties of $\kappa$-Cl based on the Hubbard model, taking into account the distinct two types of dimers and the anisotropy in the transfer integrals between them[19], $(t_a, t_p, t_q, t_b) = (-0.207, -0.102, 0.043, -0.067)$ eV, evaluated by a first-principles calculation[24] (see Fig. 2b). At three-quarter filling where the number of electrons in the unit cell is equal to 6, the ground state exhibits a metal-to-insulator transition from a paramagnetic (PM) phase to an AFM phase on increasing the intra-molecular Coulomb interaction $U$[19,25].

**Spin splitting**. A crucial feature in the AFM state of $\kappa$-Cl is that up and down spins are situated on the dimers with the different orientations as shown in Fig. 2b, resulting in the glide symmetry breaking with respect to the $yz$ plane. Here we consider the glide operation not acting on the spins. The molecular orientation makes the AFM state not invariant under the combination of time reversal and spatial translation operations, unlike simple Néel-type AFM state, e.g., on the square lattice. This situation

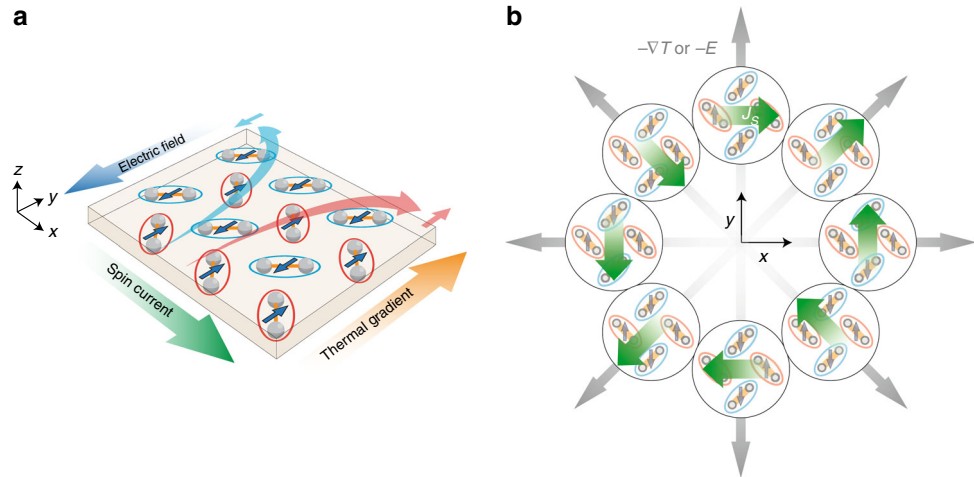

**Fig. 1** Schematic illustrations of the spin current generation. **a** Flows of up- and down-spin magnons (electrons) and spin current driven by a thermal gradient (an electric field) in the AFM state. The red and blue ellipses represent the two kinds of molecular dimers, forming a checker-plate-type lattice. The arrows in the dimers represent the localized spin moments. **b** Field-angle dependence of the spin current generation (green arrows) driven by a thermal gradient or an electric field (gray arrows)

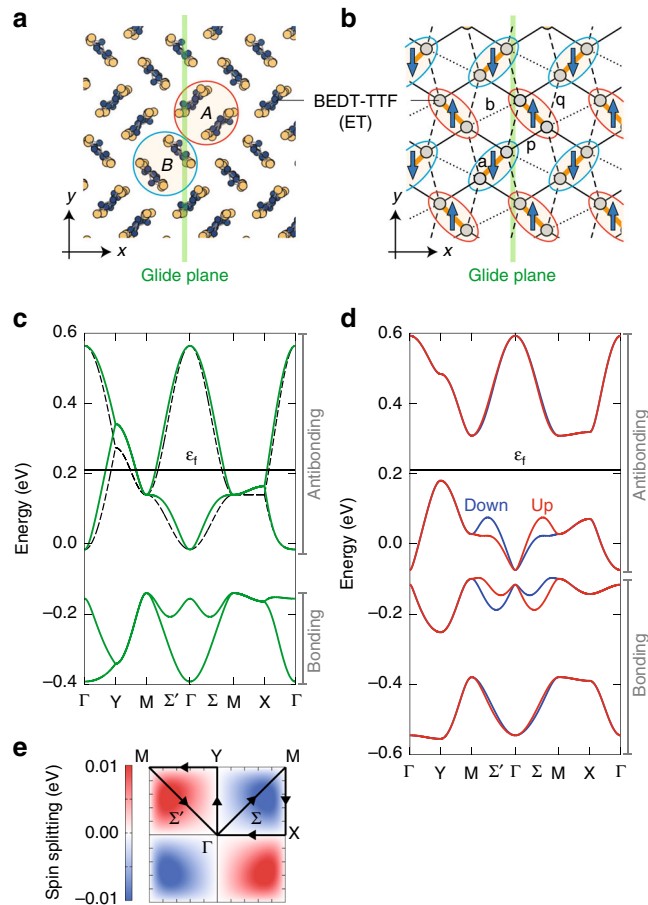

**Fig. 2** Schematic illustration of the lattice structure of $\kappa$-Cl and the energy bands. **a** Molecular arrangement in the two-dimensional conducting layer. The red and blue circles represent the two kinds of ET dimers, termed $A$ and $B$, respectively, in the unit cell. The green line denotes the glide plane perpendicular to the $xy$ plane. **b** Network of the dominant electron transfer bonds, a (orange bold line), b (dotted line), p (solid line), and q (broken line). The gray circles represent the ET molecules, and the red and blue ellipses show the $A$ and $B$ dimers, respectively. The arrows represent the local spin moments in the AFM phase. We note that another glide plane exists when considering the layer stacking, but it does not affect our discussions. **c** Energy band dispersion composed of the frontier orbitals of ET molecules in the PM metallic phase with the transfer integrals $(t_a, t_p, t_q, t_b) = (-0.207, -0.102, 0.043, -0.067)$ eV (green solid line) and that of the single-band picture in the large dimerization limit (broken line). The average electron number in the unit cell is 6 and the Fermi energy $\varepsilon_f$ is shown. **d** Energy band dispersion in the AFM insulating phase with the intra-molecular Coulomb interaction $U = 1$ eV, within the self-consistent mean-field theory. **e** Contour map of the spin splitting subtracting the down-spin energy from the up-spin energy of the top band in **d** in the first Brillouin zone. The trajectory shows the symmetric lines in **c** and **d**

gives rise to an energy band splitting depending on the spins, which has been overlooked previously. Figure 2d shows the band structure in the AFM state, calculated within the self-consistent mean-field theory (see Methods). The spin splitting appears in the whole Brillouin zone except on the $k_x$-, $k_y$-axes and the zone boundary as shown in Fig. 2e.

The origin of the spin splitting is understood from the real-space anisotropy induced by the AFM ordering as follows. Figure 3 shows the effective inter-dimer transfer integrals between the antibonding orbitals, calculated by the second-order perturbation with respect to the inter-orbital hybridizations

(see Methods). In the PM phase, as shown in Fig. 3a, b, the $A$ and $B$ dimers show different real-space anisotropies owing to the molecular orientations, but the anisotropies are symmetric with respect to the glide operation and do not depend on the spin degree of freedom. In the AFM phase, in contrast, the transfer integrals for up-spin electrons on the $A$ dimer (Fig. 3c) and down-spin electrons on the $B$ dimer (Fig. 3f) are enhanced, whereas their counterparts (Fig. 3d, e) are reduced. This spin-dependent anisotropy leads to the spin splitting.

The real-space anisotropies also show up in the effective spin exchange interactions in the Heisenberg model, derived from the above Hubbard model. Note that the system retains $SU(2)$ symmetry because of the absence of the spin-orbit coupling. Figure 4a shows the spatial distributions of the nearest-neighbor (NN) exchange interactions $J$ and $J'$, and the next-nearest-neighbor (NNN) interactions $K$ and $K'$. $K$ and $K'$ arise from fourth-order perturbation processes with respect to the NN transfer integrals (see Methods). As shown in Fig. 4b, $K'$ becomes much smaller than $K$ for realistic parameters. Then, the AFM magnon dispersion of the Heisenberg model exhibits a spin splitting as shown in Fig. 4c, where we take $K = 2$ meV and $K' = 0$ for simplicity, and $J = 80$ meV and $J' = 20$ meV[14] (see Methods). Similar spin splitting was reported in non-centrosymmetric systems with the spin-orbit coupling[26,27], but the present mechanism requires neither non-centrosymmetry nor the spin-orbit coupling.

**Spin current by a thermal gradient**. The spin-split magnon excitations lead to a spin current generation. Figure 4d shows the off-diagonal spin current conductivity, along the $x$-axis with respect to the thermal gradient along the $y$-axis, $\chi_{xy}^{SQ}$, as a function of temperature $T$ and the exchange interaction $K$, calculated by the linear response theory (see Methods). The range of $T$ is chosen well below the Néel temperature of $\kappa$-Cl, 23 K. The polarization of the spin current is parallel to the AFM moment, and the damping factor $\eta$ is fixed at 1 meV. We obtain nonzero $\chi_{xy}^{SQ}$ for $T>0$ and $K>0$, which monotonically increases in proportion to $T^2$ and $K$. Remarkably, the conductivity tensor $\chi^{SQ}$ is symmetric, $\chi_{xy}^{SQ} = \chi_{yx}^{SQ}$, with vanishing diagonal elements, $\chi_{xx}^{SQ} = \chi_{yy}^{SQ} = 0$. This leads to the peculiar field-angle dependence that we showed in Fig. 1b, which is distinct from the conventional spin Nernst effect where the spin current is always perpendicular to the thermal gradient.

This spin current generation is a direct consequence of the magnon dispersion in Fig. 4c which indicates that the up-spin magnon has high mobility along the $(1,1)$ and $(-1,-1)$ directions, while the down-spin magnon has along the $(1,-1)$ and $(-1,1)$ directions. When the temperature gradient is applied along the $y$-axis as shown in Fig. 1a, the up- and down-spin magnons are rectified toward the $(1,1)$ and $(1,-1)$ directions, respectively, in a symmetric way. Accordingly, a pure spin current, where the net magnon current is canceled out, is generated along the $x$-axis. This gives rise to the positive transverse $\chi^{SQ}$ in Fig. 4d. On the other hand, if the temperature gradient is parallel to the $(1,1)$ direction, while the transverse component disappears, a net up-spin magnon current is generated parallel to the field as a result of the incomplete cancellation (see Fig. 1b). This provides a finite longitudinal component of $\chi^{SQ}$ in the rotated coordinate, which is consistent with the symmetric form of the conductivity tensor.

We find that $\chi_{xy}^{SQ}$ is inversely proportional to the damping factor $\eta$ and diverges in the clean limit ($\eta = 0$), in analogy with the diagonal thermal conductivities $\kappa_{xx}$ and $\kappa_{yy}$ (see Supplementary Fig. 1a). This indicates that the ratio $\alpha \equiv |2J\chi_{\mu\nu}^{SQ}/\hbar\kappa_{\nu\nu}|$, which

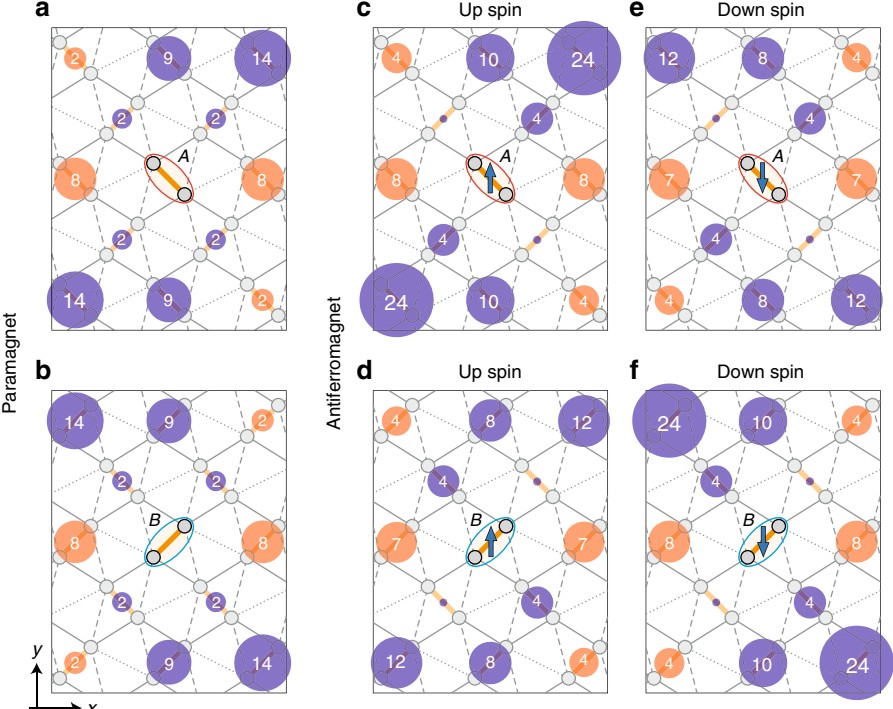

**Fig. 3** Spatial anisotropy of inter-dimer transfer integrals. **a**, **b** Effective transfer integrals between the central dimer (*A* in **a** and *B* in **b**) and the surrounding ones, obtained from the second-order perturbation processes with respect to the bonding-antibonding orbital hybridizations, in the PM phase. **c–f** Effective transfer integrals calculated likewise for up- (**c**, **d**) and down-spin (**e**, **f**) electrons in the AFM phase (the local magnetic moment is about 0.168 $\hbar$). The areas of the red (blue) shaded circles represent the amplitudes of positive (negative) transfer integrals. The amplitudes are shown in the circles in unit of meV

is used in the literatures as the conversion rate from the heat-current to the spin current, does not depend on $\eta$, however, depends on the field angle. Therefore, we choose $\mu = x$ and $\nu = y$ since $\kappa_{\nu\nu}$ is largest in this direction, considering its implication as the conversion rate. Figure 4e shows $K$ dependences of $\alpha$ at $k_B T = 0.5$ meV and 1 meV linearly increasing with $K$, but almost independent of $T$. The heat-spin current conversion efficiency reaches $\sim 5\%$ for the case of $\kappa$-Cl, which is close to one-quarter of that in Pt due to the strong spin-orbit coupling[28].

**Spin current by an electric field**. Now we propose another way of a spin current generation, in carrier doped metallic regions. The carrier doping has recently been realized experimentally[17,18]. We here focus on the electron-doping case where the AFM metallic state is stable in our model. Figure 5a shows the off-diagonal spin current conductivity induced by the electric field, $\chi_{xy}^{SC}(= \chi_{yx}^{SC})$, as a function of the Coulomb interaction $U$ and the number of electrons in the unit cell $n$ in the ground state (see Methods). $\chi_{xy}^{SC}$ is zero in the PM metallic and the AFM insulating phases, while it turns finite in the AFM metallic phase where the Fermi energy lies in the top band in Fig. 2d, whose spin splitting was shown in Fig. 2e. We note that the sign of $\chi_{xy}^{SC}$ changes around $n = 6.2$, associated with the change in the Fermi surface topology as shown in the insets of Fig. 5a. The conductivity tensor is also symmetric with zero diagonal components and inversely proportional to the damping factor (see Supplementary Fig. 1b). This means that the field-angle dependence is the same as that of $\chi_{xy}^{SQ}$ in the insulating case. It is comprehensible from the spin-dependent anisotropy of the electron transfers in Fig. 3c–e, reflected in the anisotropy in the spin-split band in Fig. 2e with the same character as the magnon band in Fig. 4c. We define the

charge-spin current conversion rate by $\beta \equiv |2e\chi_{yx}^{SC}/\hbar\sigma_{xx}|$, in analogy with $\alpha$ above (the electrical conductivity $\sigma_{\nu\nu}$ becomes largest in the $\nu = x$ direction due to the quasi-one-dimensional Fermi surfaces). As shown in Fig. 5b, in the lightly doped region with small $\sigma_{xx}$, $\beta$ becomes relatively large and approaches 7%, comparable to the spin Hall effect in Pt[29], while in the highly doped region, it decreases because of the suppression of the AFM ordering and the spin splitting.

The spin current generation in our mechanism is expected to be observed at sufficiently low temperature compared to the Néel temperature, which is not determined for doped $\kappa$-Cl. We anticipate it to be lower than the undoped case of 23 K, but to remain the same order in the lightly doped region[30].

## Discussion

The present spin current generation is strikingly different from the conventional spin Nernst and spin Hall effects. In the conventional mechanisms, a spin current is activated by the spin–orbit coupling in non-centrosymmetric lattice structures. The conductivity tensor is antisymmetric, namely, the generated spin current is always perpendicular to the applied field direction and the conversion rate is invariant under the rotation of the field. The transverse conductivity converges to a finite value in the clean limit because of the dominant inter-band contributions[7,8]. However, the strong spin–orbit coupling also disturbs the spin polarization of carriers via the spin-flipping process.

In stark contrast, the present mechanism requires neither the spin-orbit coupling nor spatial inversion symmetry breaking. The spin current conductivity is described by the symmetric tensor which results in the peculiar field-angle dependence shown in Fig. 1b, and diverges in the clean limit due to the intra-band contributions. In $\kappa$-type ET systems, the Dzyaloshinskii-Moriya

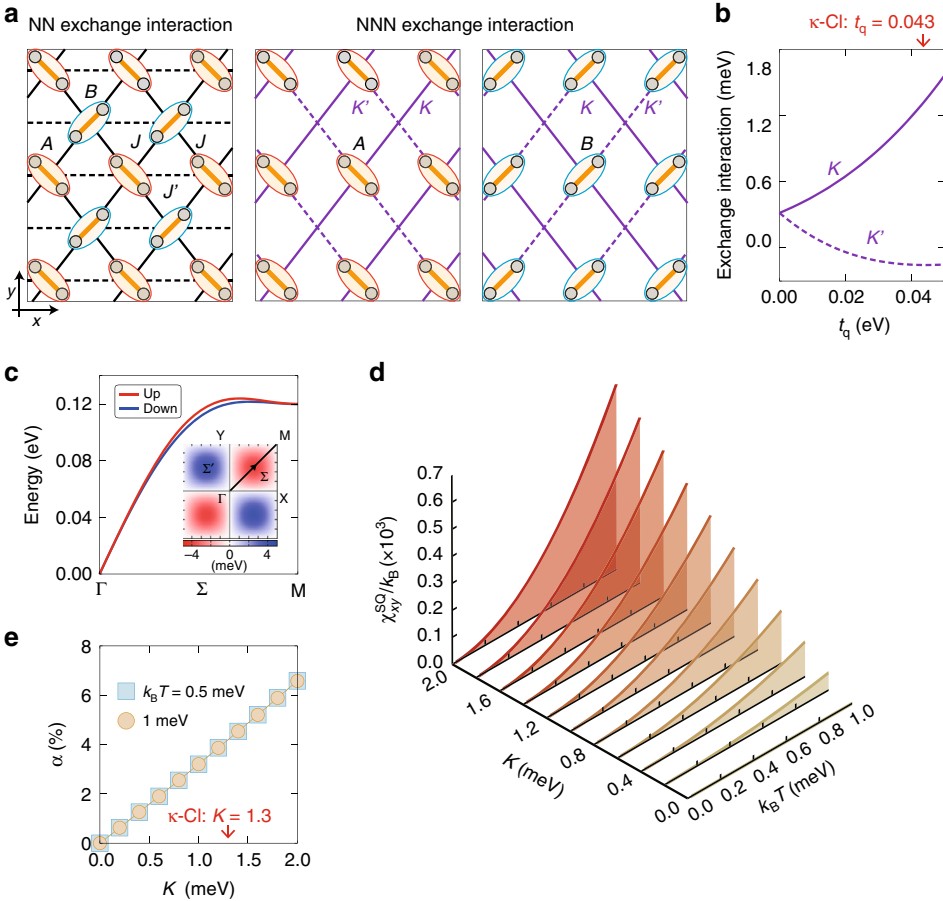

**Fig. 4 Effective NNN exchange interactions, magnon dispersions, and heat-spin current conversion in the AFM insulating state. a** Real-space distribution of the NN exchange interactions $J$ (solid lines) and $J'$ (broken lines) between $A$ and $B$ dimers (left panel) and those of the NNN exchange interactions $K$ (purple solid lines) and $K'$ (purple broken lines) between $A$ dimers (middle panel) and $B$ dimers (right panel). **b**, $t_q$ dependences of $K$ and $K'$ at $U = 1\,\text{eV}$. The red arrow represents the value of $t_q$ in $\kappa$-Cl. **c** Magnon dispersions at $(J, J', K, K') = (80, 20, 2, 0)$ meV within the linear spin-wave theory. The inset shows a contour map of the spin splitting between the up- and down-spin magnons in the first Brillouin zone. **d** $(T, K)$ dependences of the spin current conductivity under a thermal gradient, $\chi_{xy}^{SQ}$. The other exchange interactions and the damping factor are $(J, J', K', \eta) = (80, 20, 0, 1)$ meV. **e** $K$ dependences of the heat-spin current conversion rate $\alpha (= |2J\chi_{xy}^{SQ}/\hbar\kappa_{yy}|)$ at $k_B T = 0.5$ meV and 1 meV, where $\kappa_{yy}$ is the thermal conductivity along the $y$-axis. The red arrow represents the value of $K$ in $\kappa$-Cl

interaction due to the spin-orbit coupling is estimated to be a few Kelvin[15,31], which is much smaller than the NNN exchange interaction $K$. Furthermore, this class of organic charge transfer salts is known to have relatively less impurities and lattice disorders compared to inorganic crystals and organic polymers. Indeed, the specific heat and thermal transport measurements[32,33] suggest that the low temperature properties are well described by intrinsic contributions from electronic charge and spin degrees of freedom. These facts ensure a long spin lifetime in $\kappa$-Cl, which facilitates the experimental detection. Although the phenomenon has a similarity with the spin current generation in ferromagnetic metals in the sense that the time reversal symmetry is lost, the net magnetization is absent in our system; this enables us to generate a pure spin current in contrast to the spin-polarized current in ferromagnets and has the advantage of small field leakage as discussed in AFM spintronics. These considerations lead us to conclude that our proposal provides a new type of spin current generation essentially distinct from the other existing mechanisms.

As a recent experimental progress relevant to our proposal, the three-dimensional AFM structures in several $\kappa$-type ET systems have been determined by the detailed analyses of the magnetization processes[15]. It was found that $\kappa$-Cl and $\kappa$-Br show the same intra-layer AFM structure as shown in Fig. 2b, but different inter-layer stackings; the "in-phase" stacking, where the inter-layer NN spins are ferromagnetically aligned, is realized in $\kappa$-Cl, while $\kappa$-Br shows the "anti-phase" stacking. This difference will give an effective way to verify the present spin current generation because in our mechanism the sign of generated spin current is reversed by the reversal of the AFM moment. This means that a net spin current is expected in $\kappa$-Cl while it will be canceled out in $\kappa$-Br. In addition, our mechanism also has the inverse effect similar to the inverse spin Nernst or spin Hall effect, i.e., the generation of a thermal gradient or an electrical voltage by spin current injection parallel to the AFM ordered ET layers, which will give another experimental approach.

The present spin current generation arises from AFM ordering in spatially-oriented molecular orbitals. The molecular orbital degrees of freedom are fundamental and ubiquitous in organic materials. Meanwhile, similar orbital degrees of freedom are also found in inorganic materials, such as transition metal and rare-earth compounds. Thus, our new mechanism can be applied to a wide range of AFM materials. In this perspective, therefore, our finding strikes out a new direction of materials exploration for spintronics without relying on the spin-orbit coupling.

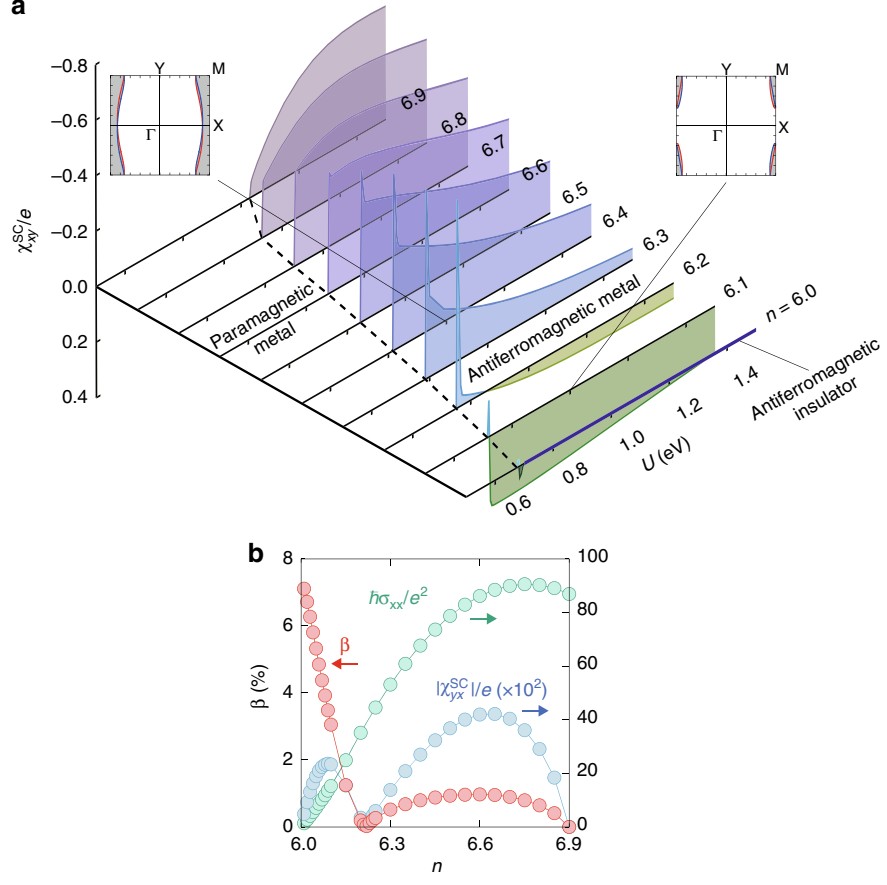

**Fig. 5** Charge-spin current conversion in the electron-doped AFM metallic state. **a** $(U, n)$ dependences of the spin current conductivity to an electric field, $\chi_{xy}^{SC}$. The broken line represents the phase boundary between the PM and AFM metallic phases, and the blue thick line shows the AFM insulating phase at three-quarter filling. The damping factor is $\gamma = 1$ meV. The insets show the Fermi surface structures of up-spin (red) and down-spin (blue) electrons at $(U, n) = (1 \text{ eV}, 6.1)$ and $(1 \text{ eV}, 6.4)$. The gray shaded areas denote the occupied states. **b** $n$ dependences of the charge-spin current conversion rate $\beta(= |2e\chi_{yx}^{SC}/\hbar\sigma_{xx}|)$, $|\chi_{yx}^{SC}|$, and the diagonal electrical conductivity $\sigma_{xx}$ at $U = 1$ eV

## Methods

**Mean-field approximation**. The Hamiltonian of the Hubbard model on the $\kappa$-type lattice is given by

$$\mathcal{H}_{\text{Hubb}} = U\sum_{i\mu} n_{i\mu\uparrow} n_{i\mu\downarrow} + t_a \sum_{i\sigma} (c_{ia\sigma}^{\dagger} c_{ib\sigma} + \text{H.c.}) + \sum_{\langle ij\rangle\mu\mu'\sigma} t_{ij}^{\mu\mu'} (c_{i\mu\sigma}^{\dagger} c_{j\mu'\sigma} + \text{H.c.}),$$

$$(1)$$

where $c_{i\mu\sigma}$ and $n_{i\mu\sigma}(= c_{i\mu\sigma}^{\dagger} c_{i\mu\sigma})$ are the annihilation operator and the number operator of an electron with a spin $\sigma(=\uparrow, \downarrow)$, on the frontier orbital of molecular site $\mu(= a, b)$ in the $i$th dimer, respectively, $U$ is the intra-molecular Coulomb interaction, and $t_a$ and $t_{ij}^{\mu\mu'}$ are the inter-molecular transfer integrals shown in Fig. 2b. We treat the Coulomb interaction term within the mean-field approximation as $n_{i\mu\uparrow} n_{i\mu\downarrow} \simeq n_{i\mu\uparrow}\langle n_{i\mu\downarrow}\rangle + \langle n_{i\mu\uparrow}\rangle n_{i\mu\downarrow} - \langle n_{i\mu\uparrow}\rangle\langle n_{i\mu\downarrow}\rangle$, and determine the expectation values self-consistently so as to minimize the total energy of the system.

**Effective electron transfer integrals**. We divide the mean-field Hamiltonian in the AFM phase into three terms as $\mathcal{H}_{\text{MF}} = \mathcal{H}_{\text{intra}} + \mathcal{H}_{\text{inter}} + \mathcal{H}_{\text{AFM}}$, where the first and second terms represent the intra-orbital and inter-orbital transfer integrals, respectively, and the third term is the local AFM field. By taking the linear combinations of the original electron operators, we define the annihilation operator of an electron in the antibonding (bonding) orbital on the $i$th dimer as $\tilde{c}_{i\alpha(\beta)\sigma} = (c_{ia\sigma} - (+)c_{ib\sigma})/\sqrt{2}$, and the three terms are given by

$$\mathcal{H}_{\text{intra}} = t_a \sum_{i\sigma} (\tilde{n}_{i\beta\sigma} - \tilde{n}_{i\alpha\sigma}) + \sum_{\langle ij\rangle\nu\sigma} (\tau_{ij}^{\nu\nu} \tilde{c}_{i\nu\sigma}^{\dagger} \tilde{c}_{j\nu\sigma} + \text{H.c.}), \quad (2)$$

$$\mathcal{H}_{\text{inter}} = \sum_{\langle ij\rangle\nu\sigma} (\tau_{ij}^{\nu\bar{\nu}} \tilde{c}_{i\nu\sigma}^{\dagger} \tilde{c}_{j\bar{\nu}\sigma} + \text{H.c.}), \quad (3)$$

$$\mathcal{H}_{\text{AFM}} = \frac{U\delta}{4} \Big(\sum_{i(\in B)\nu\sigma} - \sum_{i(\in A)\nu\sigma}\Big)\sigma\tilde{n}_{i\nu\sigma}, \quad (4)$$

where the number operator is given by $\tilde{n}_{i\nu\sigma} = \tilde{c}_{i\nu\sigma}^{\dagger} \tilde{c}_{i\nu\sigma}$, and $\bar{\nu} = \beta (\alpha)$ for $\nu = \alpha (\beta)$. The transfer integral between the neighboring dimers is given by $\tau_{ij} = \mathcal{U} t_{ij} \mathcal{U}^{\text{T}}$, by

using the two-by-two unitary matrix $\mathcal{U}$ satisfying $(\tilde{c}_{\alpha\sigma}, \tilde{c}_{\beta\sigma})^{\text{T}} = \mathcal{U}(c_{a\sigma}, c_{b\sigma})^{\text{T}}$. The amplitude of the local AFM field is given by $\delta = \langle\tilde{n}_{i\in A\uparrow}\rangle - \langle\tilde{n}_{i\in A\downarrow}\rangle = \langle\tilde{n}_{i\in B\downarrow}\rangle - \langle\tilde{n}_{i\in B\uparrow}\rangle$, where $\tilde{n}_{i\sigma} = \sum_{\nu}\tilde{n}_{i\nu\sigma}$.

We treat $\mathcal{H}_{\text{inter}}$ as the perturbation term and calculate the effective transfer integrals for the bonding and antibonding orbitals up to $\mathcal{O}(\mathcal{H}_{\text{inter}}^2)$. In the **k** space, the mean-field Hamiltonian is described by the matrix form as $\mathcal{H}_{\text{MF}} = \sum_{\mathbf{k}\sigma} \mathbf{d}_{\mathbf{k}\sigma}^{\dagger} (H_{\mathbf{k}\sigma}^{(0)} + V_{\mathbf{k}\sigma})\mathbf{d}_{\mathbf{k}\sigma}$, where $H_{\mathbf{k}\sigma}^{(0)}$ and $V_{\mathbf{k}\sigma}$ are the unperturbed and perturbed terms, respectively, given by $4\times 4$ matrices. $\mathbf{d}_{\mathbf{k}\sigma}$ is the vector of the annihilation operators of the Bloch states, which is chosen so as to diagonalize the unperturbed term as $\hat{H}_{\mathbf{k}\sigma}^{(0)}|\mathbf{k}\nu_{\sigma}^{\xi}\rangle = \varepsilon_{\mathbf{k}\nu_{\sigma}}^{\xi}|\mathbf{k}\nu_{\sigma}^{\xi}\rangle$, where $\xi(= 1, 2)$ indicates the two bands in the bonding and antibonding bands each originating from the two dimers in the unit cell. The second-order perturbation term $H_{\mathbf{k}\sigma}^{(2)}$ is decomposed into two $2\times 2$ matrices for the antibonding ($\alpha$) and bonding bands ($\beta$) as $H_{\mathbf{k}\sigma}^{(2)} = h_{\mathbf{k}\alpha\sigma}^{(2)} \oplus h_{\mathbf{k}\beta\sigma}^{(2)}$. The matrix element of $h_{\mathbf{k}\nu\sigma}^{(2)}$ is given by

$$h_{\nu;\xi\xi'}^{(2)} = \sum_{\eta=1,2} \frac{\langle\nu^{\xi}|\hat{V}|\bar{\nu}^{\eta}\rangle\langle\bar{\nu}^{\eta}|\hat{V}|\nu^{\xi'}\rangle}{\varepsilon_{\nu}^{\xi} - \varepsilon_{\bar{\nu}}^{\eta}}, \quad (5)$$

where the indices **k** and $\sigma$ are omitted for simplicity. By the Fourier transformation of $H_{\mathbf{k}\sigma}^{(2)}$, we obtain the effective transfer integrals shown in Fig. 3.

**Next-nearest-neighbor exchange interactions**. From the Hubbard model in Eq. (1), we derive the effective NNN exchange interaction in the restricted space where each antibonding orbital is occupied by one hole due to the strong Coulomb interaction $U$. The NNN exchange interaction is derived from the fourth-order perturbation process with respect to the inter-dimer transfer integrals, which is given by

$$\mathcal{H}^{(4)} = \mathcal{P}\mathcal{V}\left(\frac{1}{E_I - \mathcal{H}^{(0)}}\mathcal{Q}\mathcal{V}\right)^3 |I\rangle\langle I|, \quad (6)$$

where $\mathcal{P}$ and $\mathcal{Q}$ are the projection operators onto inside and outside of the restricted space, respectively, and satisfy $\mathcal{P} + \mathcal{Q} = 1$, $\mathcal{V}$ is the perturbation given by the third term in Eq. (1), $\mathcal{H}^{(0)}$ is the unperturbed Hamiltonian given by the first and second terms in Eq. (1), and $E_I$ is the energy of the initial eigenstate $|I\rangle$ of $\mathcal{H}^{(0)}$. The resultant exchange interaction on the NNN bond between the $A$ dimers denoted by $K$ in the middle panel of Fig. 4a is given by

$$\mathcal{H}_{ijk} = \tilde{J}\left(\mathbf{S}_i \cdot \mathbf{S}_j + \mathbf{S}_j \cdot \mathbf{S}_k - \frac{1}{2}\right) + K\left(\mathbf{S}_i \cdot \mathbf{S}_k - \frac{1}{4}\right), \quad (7)$$

where the indices $ijk$ denote the three neighboring dimers, $\mathbf{S}_i$ is the spin operator of the $i$th dimer, and $K$ is the NNN exchange constant. $\tilde{J}$ is the NN exchange constant arising from the fourth-order perturbation process, which does not contribute to the magnon splitting. The details of the fourth-order process and the explicit form of $K$ (and $K'$) are given in Supplementary Note 2.

**Linear spin-wave approximation.** The effective Heisenberg model involving the NNN exchange interaction is given by

$$\mathcal{H}_{\text{Heis}} = J\sum_{\langle ij\rangle}\mathbf{S}_i \cdot \mathbf{S}_j + J'\sum_{\langle ij\rangle'}\mathbf{S}_i \cdot \mathbf{S}_j + K\sum_{\langle\langle ij\rangle\rangle}\mathbf{S}_i \cdot \mathbf{S}_j + K'\sum_{\langle\langle ij\rangle\rangle'}\mathbf{S}_i \cdot \mathbf{S}_j, \quad (8)$$

where $\langle ij\rangle$ and $\langle ij\rangle'$ stand for the diagonal and horizontal NN bonds on the equilateral triangular lattice, $\langle\langle ij\rangle\rangle$ and $\langle\langle ij\rangle\rangle'$ are the NNN bonds shown in Fig. 4a. By using the Holstein-Primakoff transformation, we obtain the linear spin-wave Hamiltonian given by

$$\mathcal{H}_{\text{Heis}} \simeq \mathcal{H}_{\text{LSW}} = \frac{1}{2}\sum_{\mathbf{k}}\left[A_{\mathbf{k}}a_{\mathbf{k}}^{\dagger}a_{\mathbf{k}} + B_{\mathbf{k}}b_{-\mathbf{k}}^{\dagger}b_{-\mathbf{k}} + C_{\mathbf{k}}(a_{\mathbf{k}}^{\dagger}b_{-\mathbf{k}}^{\dagger} + a_{\mathbf{k}}b_{-\mathbf{k}})\right], \quad (9)$$

where $a_{\mathbf{k}}$ and $b_{\mathbf{k}}$ are the Fourier transforms of the annihilation operators of magnons on the $A$ and $B$ dimers, respectively. The coefficients are given by

$$A_{\mathbf{k}} = 4J + 2J'[\cos(\mathbf{k}\cdot\mathbf{a}_x) - 1] + 2K[\cos(\mathbf{k}\cdot(\mathbf{a}_x + \mathbf{a}_y)) - 1] + 2K'[\cos(\mathbf{k}\cdot(\mathbf{a}_x - \mathbf{a}_y)) - 1], \quad (10)$$

$$B_{\mathbf{k}} = 4J + 2J'[\cos(\mathbf{k}\cdot\mathbf{a}_x) - 1] + 2K[\cos(\mathbf{k}\cdot(\mathbf{a}_x - \mathbf{a}_y)) - 1] + 2K'[\cos(\mathbf{k}\cdot(\mathbf{a}_x + \mathbf{a}_y)) - 1], \quad (11)$$

and

$$C_{\mathbf{k}} = 2J\left[\cos(\mathbf{k}\cdot(\mathbf{a}_x + \mathbf{a}_y)/2) + \cos(\mathbf{k}\cdot(\mathbf{a}_x - \mathbf{a}_y)/2)\right], \quad (12)$$

where $\mathbf{a}_x$ and $\mathbf{a}_y$ are the primitive translational vectors. $\mathcal{H}_{\text{LSW}}$ is easily diagonalized by the standard Bogoliubov transformation, and the magnon energy dispersion shown in Fig. 4c is obtained.

**Spin current conductivity to a thermal gradient.** The spin current and energy current operators in the magnon system[34] are given by

$$\mathbf{J}_{S^z} = \frac{1}{i\hbar}[\mathbf{P}_{S^z}, \mathcal{H}_{\text{LSW}}] \quad (13)$$

and

$$\mathbf{J}_E = \frac{1}{i\hbar}[\mathbf{P}_E, \mathcal{H}_{\text{LSW}}], \quad (14)$$

respectively. $\mathbf{P}_{S^z}$ and $\mathbf{P}_E$ are the spin polarization and the energy polarization operators defined by $\mathbf{P}_{S^z} = \hbar\sum_i S_i^z \mathbf{R}_i$ and $\mathbf{P}_E = \sum_i \mathcal{H}_i \mathbf{R}_i$, respectively, where $\mathbf{R}_i$ is the position vector of the center of the $i$th dimer and $\mathcal{H}_i$ is the local energy density defined by $\mathcal{H}_{\text{LSW}} = \sum_i \mathcal{H}_i$, by the Fourier transformation of Eq. (9). In the magnon system where the chemical potential is zero, the heat-current operator $\mathbf{J}_Q$ is identical to the energy current operator $\mathbf{J}_E$. We note that the spin is a conserved quantity and the spin current is well defined here since our model does not include the spin-orbit coupling. In the linear response theory, the spin current conductivity to a static thermal gradient is given by

$$T\chi_{\mu\nu}^{\text{SQ}} = \lim_{\omega \to 0}\frac{Q_{\mu\nu}^{\text{SQ}}(\omega) - Q_{\mu\nu}^{\text{SQ}}(0)}{i\omega}, \quad (15)$$

where $\mu$ and $\nu$ represent the spatial axes $x$ and $y$. The spin-current-heat-current response function $Q_{\mu\nu}^{\text{SQ}}(\omega)$ is given by the Kubo formula

$$Q_{\mu\nu}^{\text{SQ}}(\omega) = \frac{i}{\hbar V}\int_0^{\infty} dt\, e^{it(\omega + i\eta)}\langle[J_{S^z}^{\mu}(t), J_Q^{\nu}]\rangle_{\text{eq}}, \quad (16)$$

where $\mathbf{J}_{S^z}(t)$ is the Heisenberg representation of the spin current operator, $\eta$ is the damping factor, $V$ is the volume of the system, and $\langle\cdots\rangle_{\text{eq}}$ represents the thermal average under the temperature $T$.

**Spin current conductivity to an electric field.** The spin current and charge current operators are defined by

$$\mathbf{J}_{s^z} = \frac{1}{i\hbar}[\mathbf{P}_{s^z}, \mathcal{H}_{\text{MF}}] \quad (17)$$

and

$$\mathbf{J} = \frac{1}{i\hbar}[\mathbf{P}, \mathcal{H}_{\text{MF}}], \quad (18)$$

respectively. $\mathbf{P}_{s^z}$ and $\mathbf{P}$ are the spin $s^z$ polarization and the electric polarization operators defined by $\mathbf{P}_{s^z} = \hbar\sum_i s_i^z \mathbf{r}_i$ and $\mathbf{P} = -e\sum_i \mathbf{r}_i$, respectively, where $s_i^z = \frac{n_{i\uparrow} - n_{i\downarrow}}{2}$ is the spin operator of the $i$th molecule at the position vector $\mathbf{r}_i$. The spin current conductivity to an static electric field is given by

$$\chi_{\mu\nu}^{\text{SC}} = \lim_{\omega \to 0}\frac{Q_{\mu\nu}^{\text{SC}}(\omega) - Q_{\mu\nu}^{\text{SC}}(0)}{i\omega}. \quad (19)$$

The spin-current-charge-current response function $Q_{\mu\nu}^{\text{SC}}(\omega)$ is given by the Kubo formula

$$Q_{\mu\nu}^{\text{SC}}(\omega) = \frac{i}{\hbar V}\int_0^{\infty} dt\, e^{it(\omega + i\gamma)}\langle[J_{s^z}^{\mu}(t), J^{\nu}]\rangle_0, \quad (20)$$

where $\mathbf{J}_{s^z}(t)$ is the Heisenberg representation of the spin current operator, $\gamma$ is the damping factor, and $\langle\cdots\rangle_0$ represents the average with respect to the ground state.

## Data availability
Data are available from the corresponding author upon reasonable request.

## Code availability
Computer codes used in this study are available from the corresponding author upon reasonable request.

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

## Acknowledgements

This work is supported by Grant-in-Aid for Scientific Research, No. JP16K17731, No. JP19K03723, No. JP18H04296 (J-Physics), No. JP18K13488, No. JP15H05885 (J-Physics), No. JP19K03752, and No. JP26400377 from MEXT (Japan).

## Author contributions

M.N., S.H., H.K., Y.Y., Y.M., and H.S. contributed to conception, execution, and write-up this project. The numerical and analytical calculations were performed by M.N.

## Additional information

**Competing interests:** The authors declare no competing interests.

