## [Peer Review File · Nature Communications]

Reviewers' Comments:

Reviewer #1:

Remarks to the Author:

The manuscript from Naka and co-workers proposes an intriguing phenomenon, namely the possibility of observing spin currents in an organic antiferromagnet. This is a pretty interesting idea, somehow sought after in the organic spintronics world. Not having any substantial spin-orbit coupling and presenting generally low mobilities, organic compounds do not seem suitable to host spin currents. Yet, the authors came up with some clever idea, namely to use the crystal symmetry as a tool for producing spin currents. The paper is based essentially on investigating a Hubbard model at the mean field level on a particular crystal structure. The compound in discussion exists, so that one may expect an experimental verification of the theory proposed here. Overall I believe that the manuscript will deserve publication.

However, I think that a few changes/additions will improve the paper.

1) The paper presents a limited attempt to provide an intuitive picture of the physics at play here. The presentation is driven by the results of the model (obtained with a range of methodologies presented in the supplementary material), but makes little attempts to provide a simple intuitive justification of the results. I believe that some lay-man explanation of the effects presented here will help the reader.

2) Although indicated in the abstract that the currents can be generated by both a thermal gradient and an electric field, at the end of the day one finds this out only when presenting the results. In order to make the presentation more tutorial, I suggest to include a schematic picture describing the type of measurements that the authors are going to discuss. This may also help the final section, where the invariances with respect to a magnetic field are discussed and compared with the standard spin-orbit based mechanism.

3) In general antiferromagnetism in organic materials is expected always to occur at rather low temperatures. The authors should comment about that, so that the present result can be put in prospective.

4) In the metallic case can the authors comment on the robustness of the antiferromagnetic order? What is the Neel temperature in that case?

In conclusion I find this to be a nice and clever idea, that should deserve publication. However, the authors have first to amend the manuscript along the lines described above.

Reviewer #2:

Remarks to the Author:

The manuscript describes a highly innovative mechanism for generating spin-current, that does not rely on spin-orbit coupling. Instead it is based on a spin-dependent anisotropy of the electronic structure. The work is exclusively theoretical in nature. The theory appears sound, but I have to admit to not being a theorist in the field. For that reason, and more generally given its innovative nature, I would really like to see the authors back up their findings with experimental results found in the literature, even if they are only relevant to partial results and certain aspects of the work. Any reference to the literature to similar systems and ideas would enhance my confidence in this work. At present the work appears somewhat isolated.

Otherwise I will need to study the comments of other referees, and the responses of the authors to said comments, before I can firm a solid opinion and recommendation.

Reviewer #3:

Remarks to the Author:

The authors performed a theoretical investigation about the spin current generation in organic antiferromagnets (k-Cl) in the case of electron doping. They claimed a spin current can be obtained by applying a thermal gradient or an electric field, in the absence of spin-orbit interactions. The spin-charge conversion ratio is comparable to Pt. The theory is detailed and the model calculations are elaborate.

These conclusion sounds very novel and interesting, although experimental verification is still absent. The claims provide an alternative way to generate spin currents besides spin-Hall effect, especially for organics. However, I found the physical picture of the phenomenon is not elucidate clearly. Some details and possible comparison are needed, which are important in determining the significance of the work. Before answering the following questions, I will not recommend this work published in Nature Communications which has a high criterion.

(1) Is this spin current presented here a pure spin current similar as that in spin-Hall effect, or only a spin-polarized current like spin injection where the spin-resolved components are unequal?

(2) In the AFM phase, applying an electric field, e.g., E_x along x direction, could you give a clear description of such charge-spin conversion picture? Is there net charge current along y or other directions? Does the spin current exists on both x and y directions? Especially in the case of conductivity anisotropy.

(3) Is there a possible inverse process to detect the spin current?

(4) What's the advantage of the method and materials that cannot be realized easily? I mean whether these results can be easily obtained by applying an electric field on an organic ferromagnet in case of dopant. What is their difference in nature?

(5) Is there any possible "organic effects" during such process in this organic material? E.g., disorder effect in organic spin-Hall effect [Phys. Rev. Lett. 115, 026601 (2015)] or polaronic effect in organic ferromagnets [Phys. Rev. B 98, 235415 (2018)].

=== Response to Reviewer #1

Reviewer #1 (Remarks to the Author):

The manuscript from Naka and co-workers proposes an intriguing phenomenon, namely the possibility of observing spin currents in an organic antiferromagnet. This is a pretty interesting idea, somehow sought after in the organic spintronics world. Not having any substantial spin-orbit coupling and presenting generally low mobilities, organic compounds do not seem suitable to host spin currents. Yet, the authors came up with some clever idea, namely to use the crystal symmetry as a tool for producing spin currents. The paper is based essentially on investigating a Hubbard model at the mean field level on a particular crystal structure. The compound in discussion exists, so that one may expect an experimental verification of the theory proposed here. Overall I believe that the manuscript will deserve publication. However, I think that a few changes/additions will improve the paper.

We thank Reviewer #1 for appreciating our study by a very positive opinion recommending the publication in Nature Communications, and for helpful comments to improve our paper. We have revised the manuscript following the reviewer's suggestions. Below are the responses to the comments: since 1), 2) and 3), 4) are related, respectively, we respond to them together.

1) The paper presents a limited attempt to provide an intuitive picture of the physics at play here. The presentation is driven by the results of the model (obtained with a range of methodologies presented in the supplementary material), but makes little attempts to provide a simple intuitive justification of the results. I believe that some lay-man explanation of the effects presented here will help the reader.

2) Although indicated in the abstract that the currents can be generated by both a thermal gradient and an electric field, at the end of the day one finds this out only when presenting the results. In order to make the presentation more tutorial, I suggest to include a schematic picture describing the type of measurements that the authors are going to discuss. This may also help the final section, where the invariances with respect to a magnetic field are discussed and compared with the standard spin-orbit based mechanism.

We thank the reviewer for a valuable suggestion. We have added a schematic figure and a brief introduction of the present mechanism of spin current generation in the first section in the revised manuscript [see the summary of changes (1) and (11)]. This will give an intuitive picture of the physics as well as the type of measurements we are going to discuss. By inserting these in the early part of our paper, it indeed looks much better, especially for the readers with broad backgrounds.

3) In general antiferromagnetism in organic materials is expected always to occur at rather low temperatures. The authors should comment about that, so that the present result can be put in prospective.

4) In the metallic case can the authors comment on the robustness of the antiferromagnetic order? What is the Neel temperature in that case?

As described in lines 159-160 in the previous manuscript (lines 174-175 in the revised one), the Neel temperature of κ -(BEDT-TTF)₂Cu[N(CN)₂]Cl is around 23 K. The Neel temperature of the doped κ -(BEDT-TTF)₂Cu[N(CN)₂]Cl has not been reported yet because of the difficulty in magnetic measurements in thin film samples for which the doping was achieved. Nevertheless, we naturally expect that it decreases as increasing the doping rate, as widely observed in other doped antiferromagnets like high- T_c cuprates. Indeed, according to our theoretical results, the antiferromagnetic moment, which is an indicator of the amplitude of the Neel temperature, gradually decreases with increasing the doping rate, and such a tendency is also obtained by a recent numerical calculation taking into account of strong correlation effects beyond the mean field theory [H. Watanabe, H. Seo, and S. Yunoki, arXiv:1811.09035 (2018)]. Based on these, we expect that the Neel temperature is decreased by doping but remain the same order in the lightly doped region.

We have added sentences and a reference about the Neel temperature in the antiferromagnetic phase in the revised manuscript [see the summary of changes (3) and (9)].

In conclusion I find this to be a nice and clever idea, that should deserve publication. However, the authors have first to amend the manuscript along the lines described above.

We hope that the revisions we made meet the reviewer's request.

=== Response to Reviewer #2

Reviewer #2 (Remarks to the Author):

The manuscript describes a highly innovative mechanism for generating spin-current, that does not rely on spin-orbit coupling. Instead it is based on a spin-dependent anisotropy of the electronic structure. The work is exclusively theoretical in nature. The theory appears sound, but I have to admit to not being a theorist in the field. For that reason, and more generally given its innovative nature, I would really like to see the authors back up their findings with experimental results found in the literature, even if they are only relevant to partial results and certain aspects of the work. Any reference to the literature to similar systems and ideas would enhance my confidence in this work. At present the work appears somewhat isolated. Otherwise I will need to study the comments of other referees, and the responses of the authors to said comments, before I can form a solid opinion and recommendation.

We thank Reviewer #2 for reviewing our manuscript and recognizing the innovative nature of our theoretical results. We indeed agree with his/her comment about the lack of experimental information in our previous manuscript. To the best of our knowledge, there is no direct experimental measurement of our theoretical proposal at the present stage. Nonetheless, from the quantitative estimates based on the parameters in the actual materials, we consider that our proposed mechanism will be verified in experiments with a high probability, as described in the following.

First of all, we would like to emphasize that the lack of experiments directly related to our work at the present stage is because the spin splitting without spin-orbit coupling has been a long-overlooked idea. Secondly, it is difficult to detect such a spin splitting or the spin current by the standard experimental methods applicable to organic conductors. For example, a direct observation of the spin-split energy band by using the spin-resolved ARPES (angle-resolved photoemission spectroscopy), typically used in many systems, is difficult to be performed for organic charge transfer salts owing to the sample degradation by laser light or X-ray absorption. Furthermore, experimental efforts have hardly been made so far in organic conductors because of their weak spin-orbit coupling.

In light of these points, our theoretical proposal will be important for stimulating various experiments to capture these peculiar spin splitting and spin current generation. Indeed, we are planning for multifaceted approaches based on our theory to detect them by combining several optical, magnetic, and transport measurements in collaboration with experimental groups.

Nevertheless, there are several experimental developments, which can back up our finding. An important progress is the determination of the three-dimensional AFM spin structures in several κ -type BEDT-TTF compounds by detailed analyses of the magnetization curves in Ref. [14] in the previous manuscript (Ref. [15] in the revised one) by Ishikawa *et al.*. They have found that κ -Cl and

deuterated κ -Br show the same intra-layer AFM ordered structure, as shown in Fig. 1b in the previous manuscript (Fig. 2b in the revised one) and consistent with our calculations, but different inter-layer stacking patterns. In fact, this difference will give an effective way to verify our scenario of the spin current generation. This is because, based on our theory, if the ordered spin moments are reversed with fixing the external field, the direction of spin current is also reversed. This means that when the inter-layer AFM stacking is “in-phase” where the inter-layer nearest-neighbor spins are ferromagnetically aligned, the spin currents driven in all the layers have the same sign and a net spin current is generated, while the “anti-phase” stacking results in the cancellation of the spin currents. According to the experiments by Ishikawa *et al.*, the in-phase stacking structure is realized in κ -Cl, while the stacking structure in κ -Br is the anti-phase one. This fact enables us to examine our theoretical proposal directly by comparing the spin current generation phenomena in these two candidate materials.

Although we have few literatures more directly related to our proposal, the most relevant one might be the paper by Z. Qiu *et al.*, AIP Advances **5**, 057167 (2015). They have fabricated a device based on a κ -type BEDT-TTF system and performed measurements of the inverse spin Hall effect, motivated by its detection in the organic polymer PEDOT:PSS reported in Ref. [11] by Ando *et al.*. Their study clearly shows that the experimental verification of our theoretical proposal using the inverse spin Hall effect is feasible, although their sample was paramagnetic or spin-glass-like state, not antiferromagnetic. It is worth noting that our mechanism will provide a larger spin current conversion efficiency than the spin Hall effect by the spin-orbit coupling because the amplitude of the spin splitting in the energy band is expected to be about 10 meV as shown in Fig. 1e in the previous manuscript (Fig. 2e in the revised one), which is an order of magnitude larger than the spin-orbit coupling in κ -Cl estimated in Ref. [29] in the previous manuscript (Ref. [30] in the revised one).

We have added sentences and a reference about the experimental situation in the revised manuscript [see the summary of changes (6) and (8)].

=== Response to Reviewer #3

Reviewer #3 (Remarks to the Author):

The authors performed a theoretical investigation about the spin current generation in organic antiferromagnets (k-CI) in the case of electron doping. They claimed a spin current can be obtained by applying a thermal gradient or an electric field, in the absence of spin-orbit interactions. The spin-charge conversion ratio is comparable to Pt. The theory is detailed and the model calculations are elaborate. These conclusion sounds very novel and interesting, although experimental verification is still absent. The claims provide an alternative way to generate spin currents besides spin-Hall effect, especially for organics. However, I found the physical picture of the phenomenon is not elucidate clearly. Some details and possible comparison are needed, which are important in determining the significance of the work. Before answering the following questions, I will not recommend this work published in Nature Communications which has a high criterion.

We thank Reviewer #3 for reviewing our manuscript and for recognizing the novelty of our theory. We agree with the comments that we need a clearer physical picture of our proposed mechanism, which was a point raised also by Reviewer #1, therefore added a figure and sentences [see the summary of changes (1) and (11)]. Furthermore, the comparison to other organic materials was lacked and therefore added in the revised manuscript, which will help highlighting the distinctiveness of our work compared to others, and emphasize the feasibility in future experiments [see the summary of changes (4) and (6)].

Below are the responses to the reviewer's comments. We put our response to (1) and (2) together since they are related with each other.

(1) Is this spin current presented here a pure spin current similar as that in spin-Hall effect, or only a spin-polarized current like spin injection where the spin-resolved components are unequal?

(2) In the AFM phase, applying an electric field, e.g., E_x along x direction, could you give a clear description of such charge-spin conversion picture? Is there net charge current along y or other directions? Does the spin current exists on both x and y directions? Especially in the case of conductivity anisotropy.

The character of the spin current generated by our mechanism changes depending on the angle of the applied external field. In fact, the field-angle dependence is a prominent signature of the present mechanism, which will serve to experimentally separate this phenomenon from the conventional spin Hall and Nernst effects.

For example, in the metallic case, this is understood from the real-space anisotropy of the electron transfers shown in Figs. 2c-2f in the previous manuscript (Figs. 3c-3f in the revised one). These figures show that the up- and down-spin electrons have different favorable directions to move, along the (1, 1) and (-1, -1) directions and the (1, -1) and (-1, 1) directions, respectively. When the electric field is applied along the x-axis, i.e., the (1, 0) direction, the up- and down-spin electrons traveling along the (-1, 0) direction drift to the (-1, -1) and (-1, 1) directions, respectively, by this asymmetry of the transfers. Accordingly, along the (0, -1) direction, the pure spin current ($J_{\uparrow}-J_{\downarrow}$) without the net charge current ($J_{\uparrow}+J_{\downarrow}$) appears. In the same manner, when the electric field is applied along the (0, 1) direction, the pure spin current is generated along the (-1, 0) axis. These are consistent with the negative transverse χ^{SC} shown in Fig. 4a in the previous manuscript (Fig. 5b in the revised one).

On the other hand, when the electric field is applied along the (1, 1) direction, where the amplitude of up-spin electron transfer is larger than that of down-spin, the up-spin current with the net charge current appears parallel to the electric field. In other words, the generated spin current becomes a spin polarized charge current in this case.

To clarify this point, we have added explanations about the field-angle dependence in the revised manuscript, at an early stage for a clearer description of our mechanism [see the summary of changes (2)] and added a new figure in Fig. 1b with the field-angle dependence revised from Fig. 3e in the previous manuscript [see the summary of changes (11)].

(3) Is there a possible inverse process to detect the spin current?

The inverse effects are surely expected. For example, when the pure spin current is injected along the (1, 0) direction in the metallic (insulating) phases, the electrical voltage (temperature gradient) will appear along the (0, 1) direction, according to the field-angle dependence in the response above to comments (1) and (2). We have added a sentence about this [see the summary of changes (7)].

(4) What's the advantage of the method and materials that cannot be realized easily? I mean whether these results can be easily obtained by applying an electric field on an organic ferromagnet in case of dopant. What is their difference in nature?

When an electric field is applied to a ferromagnetic metal, a spin polarized charge current is generated. On the other hand, in our mechanism, the pure spin current without charge current can be obtained by properly choosing the field direction, as mentioned above. In addition, organic antiferromagnets are more common compared to organic ferromagnets, which will widely expand the material foundation for spintronics research. We have added a sentence to emphasize this point [see the summary of changes (5)].

(5) Is there any possible “organic effects” during such process in this organic material? E.g., disorder effect in organic spin-Hall effect [Phys. Rev. Lett. 115, 026601 (2015)] or polaronic effect in organic ferromagnets [Phys. Rev. B 98, 235415 (2018)].

In these papers that the reviewer pointed out, the authors basically discuss spin current transport phenomena in organic polymers. On the other hand, the target system of our study is the charge transfer salts of small molecules forming single crystals, which have relatively less lattice disorder and defects compared to polymer systems. In particular, in the BEDT-TTF systems discussed in our work, based on several experimental results such as the specific heat [e.g., S. Yamashita *et al.*, Nature Physics **4**, 459 (2008)] and the thermal transport [e.g., M. Yamashita *et al.*, Nature Physics **5**, 44 (2009)] measurements, the low temperature properties are dominated by electronic charge and spin degrees of freedom with less contribution from extrinsic defects, and the effect of electron-lattice coupling is not so relevant.

We have added a sentence and references about this [see the summary of changes (4) and (10)].

====Summary of changes

- (1) From line 51 to 66 in page 1: we have added a brief introduction of the present spin current generation, “Figure 1a provides a schematic illustration of ...”.
- (2) From line 186 to 204 in page 2, from line 237 to 242 in page 3, and from line 276 to 277 in page 3: we have added detailed and intuitive explanations for the spin current generation, “This spin current generation is a direct ...”, “This means that the field-angle dependence is ...”, and “which results in the peculiar field-angle dependence ...”, respectively. Accordingly, we modified a sentence from line 181 to 182.
- (3) From line 252 to 257 in page 3: we have added sentences about the Neel temperature of organic antiferromagnets, “The spin current generation in our mechanism ...”.
- (4) From line 282 to 289 in page 3: we have added explanations about the impurity and lattice effects; we replaced the sentence “Furthermore, the organic compounds have relatively less impurities” in line 235-236 of page 3 in the previous manuscript by “Furthermore, this class of organic charge transfer ...”.
- (5) From line 294 to 296 in page 3: we have added a description about an advantage of the spin current generation in antiferromagnets, “this enables us to generate a pure ...”.
- (6) From line 301 to 314 in page 3: we have added sentences about the experimental situation relevant to our results, “As a recent experimental progress relevant to ...”.
- (7) From line 315 to 319 in page 3: we have added a sentence about the inverse effects, “In addition, our mechanism also has the ...”.
- (8) In ref. [12], we have added a reference about the measurement of the inverse spin Hall effect in a κ -type BEDT-TTF system.
- (9) In ref. [31], we have added a reference related to the Neel temperature of the metallic AFM phase.
- (10) In refs. [32] and [33], we have added references about the specific heat and thermal transport measurements in κ -type BEDT-TTF systems.
- (11) Fig. 1: We have added a schematic illustration of the spin current generation in Fig. 1a and moved the field-angle dependence in Fig. 3e in the previous manuscript to Fig. 1b with slight modifications. Accordingly, Figs. 1, 2, 3, and 4 in the previous manuscript are relabeled as Figs. 2, 3, 4, and 5, respectively.

All the above changes are highlighted in red in the revised manuscript. In addition, we made the following corrections.

- (12) Aside from the reviewers’ comments, we have noticed that there was a mistake in the direction of the AFM moments in Figs. 3c, 3d, and 3e in the previous manuscript, which correspond to Figs. 4c, 4d, and Fig. 1b in the revised manuscript, where these were reversed from the correct direction. We

have corrected these in the revised manuscript.

(13) For better readability, we have corrected some expressions and typos, e.g., “two dimensional”→ “two-dimensional” and “field-angular”→“field-angle”.

Reviewers' Comments:

Reviewer #1:

Remarks to the Author:

The authors have taken on board all the suggestions I made in my first round of review. I believe that the manuscript has then improved and it should be published.

Reviewer #2:

Remarks to the Author:

All the referees have commented positively on the innovative nature of the manuscript, but have raised similar concerns. The authors have responded diligently to the reviewers' concerns, but I think that none of the principal concerns have been conclusively eliminated. In particular, the nature, scope and robustness of the mechanism remain somewhat tenuous. For that reason, I will stop short of recommending the manuscript for publication.

Reviewer #3:

Remarks to the Author:

The author has answered my questions clearly and revised the manuscript in detail. I would like to recommend this work to be published due to the proposed novel phenomenon, although a possible experimental test is still absent. The two points have also been confirmed by other two reviewers. I hope this work may bring hints to experimental researchers, and trigger the related experiment recently.

=== Response to Reviewer #1

The authors have taken on board all the suggestions I made in my first round of review. I believe that the manuscript has then improved and it should be published.

We appreciate Reviewer #1 for recommending our paper for publication in Nature Communications. The manuscript has been improved thanks to the Reviewer's helpful comments and suggestions.

=== Response to Reviewer #2

All the referees have commented positively on the innovative nature of the manuscript, but have raised similar concerns. The authors have responded diligently to the reviewers' concerns, but I think that none of the principal concerns have been conclusively eliminated. In particular, the nature, scope and robustness of the mechanism remain somewhat tenuous. For that reason, I will stop short of recommending the manuscript for publication.

We thank Reviewer #2 for reviewing our manuscript. We would like to make our remark on the comment.

As explained in the previous manuscript, the essential nature of the present spin current generation mechanism is the combination of the oriented molecules and antiferromagnetic ordering. Molecular orientation is generic property in organic materials, which is also basically equivalent to anisotropic electronic orbitals seen in various inorganic materials, e.g., transition metal oxides. This indicates that our mechanism is not an accidental property limited in a particular compound but can be generalized to a wide range of organic and inorganic antiferromagnets.

To highlight these points, we have added a sentence and modified the last paragraph in the revised manuscript [see the summary of changes (1)].

=== Response to Reviewer #3

The author has answered my questions clearly and revised the manuscript in detail. I would like to recommend this work to be published due to the proposed novel phenomenon, although a possible experimental test is still absent. The two points have also been confirmed by other two reviewers. I hope this work may bring hints to experimental researchers, and trigger the related experiment recently.

We thank Reviewer #3 for recommending our manuscript for publication in Nature Communications. The manuscript has been improved thanks to the Reviewer's comments and suggestions, especially on the experimental relevance of our proposal. We agree on hoping this proposal to bring experimental

developments in different directions.

====Summary of changes

(1) From line 325 to 335 in page 3: we have added a sentence and modified descriptions, to highlight the nature, scope, and robustness of the present spin current generation, “**The present spin current generation arises from ...**”.

(2) We have corrected the manuscript format.

- References are omitted in the Abstract.
- We divided the sections and put subsection titles in the Results section.
- Following the editorial requests, formal corrections are made.